# Whole-Genome Sequencing Reveals the High Nosocomial Transmission and Antimicrobial Resistance of *Clostridioides difficile* in a Single Center in China, a Four-Year Retrospective Study

Xin Wen,[a,b] Cong Shen,[a,b,c,d] Jinyu Xia,[e] Lan-Lan Zhong,[a,b] Zhongwen Wu,[f] Mohamed Abd El-Gawad El-Sayed Ahmed,[a,b,g] Nana Long,[h] Furong Ma,[i] Guili Zhang,[a,b] Wenwei Wu,[a,b] Jianlve Luo,[a,b] Yong Xia,[i] Min Dai,[h] Liyan Zhang,[j] Kang Liao,[f] Siyuan Feng,[a,b] Cha Chen,[c,d] Yishen Chen,[k] Wenji Luo,[k] Guo-Bao Tian[a,b]

[a]Department of Microbiology, Zhongshan School of Medicine, Sun Yat-sen University, Guangzhou, China
[b]Key Laboratory of Tropical Diseases Control (Sun Yat-sen University), Ministry of Education, Guangzhou, China
[c]The Second Clinic Medical College, Guangzhou University of Chinese Medicine, Guangzhou, People's Republic of China
[d]Department of Clinical Laboratory, The Second Affiliated Hospital of Guangzhou University of Chinese Medicine, Guangdong Provincial Hospital of Traditional Chinese Medicine, Guangzhou, People's Republic of China
[e]Department of Infectious Diseases, The Fifth Affiliated Hospital, Sun Yat-Sen University, Zhuhai, China
[f]Department of Clinical Laboratory, The First Affiliated Hospital of Sun Yat-Sen University, Guangzhou, China
[g]Department of Microbiology and Immunology, Faculty of Pharmaceutical Sciences and Drug Manufacturing, Misr University for Science and Technology (MUST), Cairo, Egypt
[h]School of Laboratory Medicine, Chengdu Medical College, Chengdu, China
[i]Department of Clinical Laboratory Medicine, The Third Affiliated Hospital of Guangzhou Medical University, Guangzhou, China
[j]Department of Clinical Laboratory, Guangdong Provincial People's Hospital/Guangdong Academy of Medical Sciences, Guangzhou, China
[k]Department of Pharmacy, The Fifth Affiliated Hospital, Sun Yat-Sen University, Zhuhai, China

**ABSTRACT** *Clostridioides difficile*, which causes life-threatening diarrheal disease, presents an urgent threat to health care systems. In this study, we present a retrospective genomic and epidemiological analysis of *C. difficile* in a large teaching hospital. First, we collected 894 nonduplicate fecal samples from patients during a whole year to elucidate the *C. difficile* molecular epidemiology. We then presented a detailed description of the population structure of *C. difficile* based on 270 isolates separated between 2015 and 2020 and clarified the genetic and phenotypic features by MIC and whole-genome sequencing. We observed a high carriage rate (19.4%, 173/894) of *C. difficile* among patients in this hospital. The population structure of *C. difficile* was diverse with a total of 36 distinct STs assigned. In total, 64.8% (175/270) of the isolates were toxigenic, including four CDT-positive (*C. difficile* transferase) isolates, and 50.4% (135/268) of the isolates were multidrug-resistant. Statistically, the rates of resistance to erythromycin, moxifloxacin, and rifaximin were higher for nontoxigenic isolates. Although no vancomycin-resistant isolates were detected, the MIC for vancomycin was higher for toxigenic isolates ($P < 0.01$). The in-hospital transmission was observed, with 43.8% (110/251) of isolates being genetically linked to a prior case. However, no strong correlation was detected between the genetic linkage and epidemiological linkage. Asymptomatic colonized patients play the same role in nosocomial transmission as infected patients, raising the issue of routine screening of *C. difficile* on admission. This work provides an in-depth description of *C. difficile* in a hospital setting and paves the way for better surveillance and effective prevention of related diseases in China.

**IMPORTANCE** *Clostridioides difficile* infections (CDI) are the leading cause of healthcare-associated diarrhea and are known to be resistant to multiple antibiotics. In the past decade, *C. difficile* has emerged rapidly and has spread globally, causing great concern among American and European countries. However, research on CDI

Address correspondence to Yishen Chen, 112366523@qq.com, Wenji Luo, 1018693996@qq.com, or Guo-Bao Tian, tiangb@mail.sysu.edu.cn.

The authors declare no conflict of interest.

remains limited in China. Here, we characterized the comprehensive spectrum of *C. difficile* by whole-genome sequencing (WGS) in a Chinese hospital, showing a high detection rate among patients, diverse genome characteristics, a high level of antibiotic resistance, and an unknown nosocomial transmission risk of *C. difficile*. During the study period, two *C. difficile* transferase (CDT)-positive isolates belonging to a new multilocus sequence type (ST820) were detected, which have caused serious clinical symptoms. This work describes *C. difficile* integrally and provides new insight into *C. difficile* surveillance based on WGS in China.

**KEYWORDS** *Clostridioides difficile*, antibiotic-resistant, infection and colonization, nosocomial transmission, virulence gene variants, whole-genome sequencing

*C*lostridioides difficile*, previously known as *Clostridium difficile*, is a strictly anaerobic, spore-forming, Gram-positive bacillus, which colonizes the intestinal tract and causes gastrointestinal infections in health care settings (1). Symptoms of *C. difficile* infection (CDI) range from mild diarrhea to serious pseudomembranous colitis, and even life-threatening toxic megacolon disease (2). Because of the high lethality and relapse rate, CDIs are classified as an "urgent threat" to the health care system and cause of great concern among US and European countries (3, 4). RT027, a hypervirulent clone that causes more serious diseases, can produce 16- and 23-times higher levels of toxin A and B, respectively, compared to other ribotypes (5). Since the outbreak of epidemic clone RT027 in the early 2000s, research focused on CDI has increased rapidly especially in European and North American countries (6). In recent years, RT027 clones have been sporadically detected in China, with an RT027 isolate first being reported in 2012 in Guangzhou (7). Although no outbreaks have occurred in China to date (8), as hypervirulent strains emerge, increased focus on the molecular epidemiological surveillance of *C. difficile* is vital. Antibiotic resistance is becoming increasingly common and is causing a global health crisis (9). *C. difficile* is known to be resistant to multiple antibiotics, especially to erythromycin, clindamycin, and fluoroquinolone (10). The susceptibility of *C. difficile* to clinical therapeutic drugs, such as metronidazole, vancomycin, and rifaximin, is also declining (11–13), highlighting the need for increased antimicrobial susceptibility surveillance of clinical *C. difficile* isolates.

In China, the molecular epidemiology and antibiotic susceptibility profiles of *C. difficile* isolates are still not completely understood, especially at the genomic level. With the development of sequencing technology, whole-genome sequencing (WGS) is increasingly becoming the preferred method for studying drug-resistant pathogens in health care systems (14). Therefore, to improve our understanding of the molecular epidemiologic, genetic, and phenotypic features of *C. difficile* in China, we performed an integrated study that included analysis of the incidence, clinical information, antibiotic resistance profile, and genomic characteristics of *C. difficile* isolates in a large teaching hospital in China. This study laid the foundation for better surveillance and effective prevention of CDI in the future.

## RESULTS

**Patient characteristics.** Of the 894 nonduplicate fecal samples collected from February 1, 2019 to January 31, 2020, 173 (19.4%) isolates were cultured successfully, of which 63.6% (110/173) were identified as toxigenic *C. difficile* (TCD) (Fig. S1). The screened patients' characteristics are shown in Table S1. We found that patients carrying *C. difficile* were younger than *C. difficile* negative patients ($P < 0.05$). The isolation rate of *C. difficile* showed a declining trend with age, which was 22.8% (38/167) in patients under the age of 18, 20.6% (99/481) in patients between the ages of 18 and 60, and 14.6% (36/246) in patients older than 60 years of age (Fig. S2A). The same trend was observed with the TCD isolates (Fig. S2B). Out of the 894 fecal samples, 857 were collected from inpatients hospitalized in 24 distinct wards (Fig. S3A). The *C. difficile* isolation rates were diverse among different wards, which was higher in the division of gastroenterology, hematology, and nephrology and was lower in the division of pulmonary (Fig. S4). Common underlying diseases in the patients in this study were inflammatory bowel diseases (IBD), malignancy,

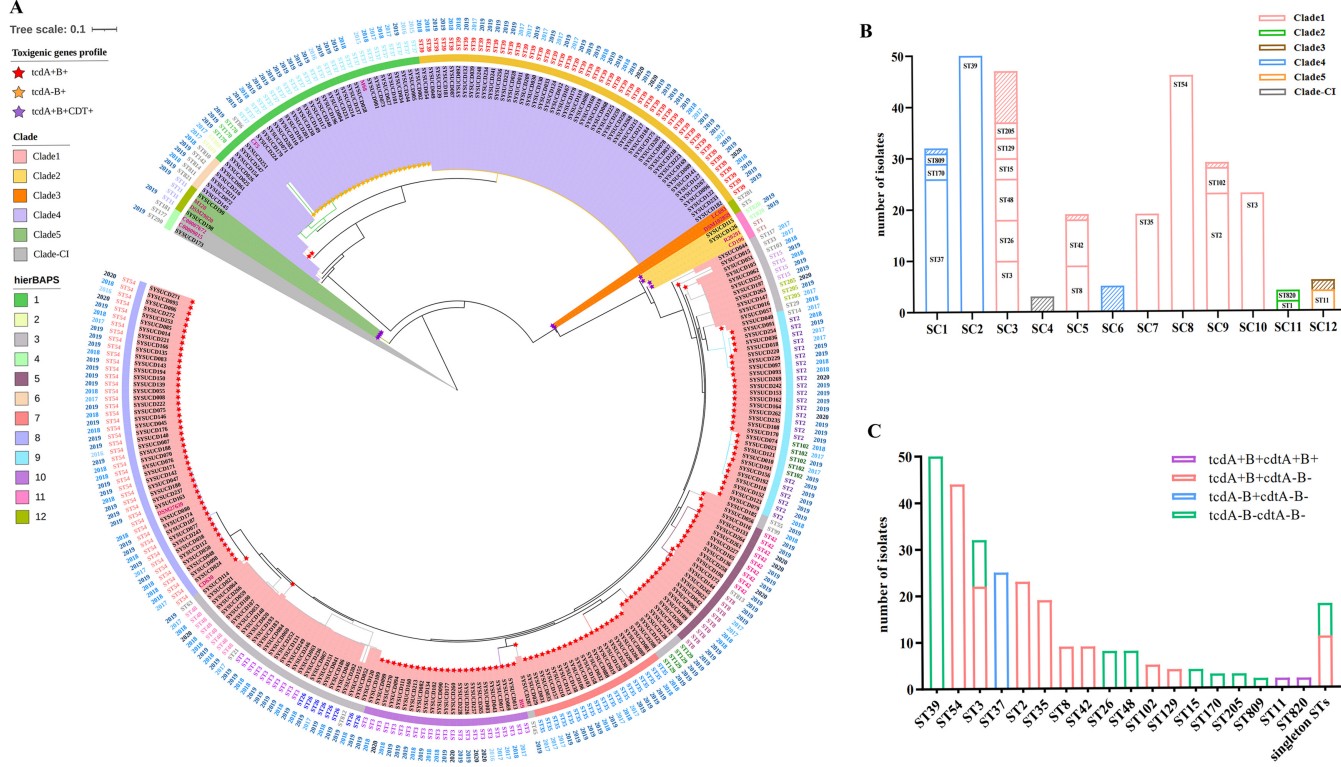

**FIG 1** Genomic structure of the *C. difficile* population. (A) A maximum likelihood phylogenetic tree of 283 *C. difficile* isolates (270 sequenced in this study and 13 reference genomes) was generated using 20057 cgSNPs from a 5.5 Mb alignment of 1529 core genes. The background shading represents the MLST clades. The internal (first) circle is colored based on sequence clusters (SCs) by hierBAPs analysis. The second circle is colored MLST data. The outside circle is the collected dates. Colored stars at the start of the branches represent different toxin profiles. (B) The correlation between three genotyping methods: clade, MLST, and SCs. The height of the bars indicates the number of isolates. Each bar represents a single SC, and the STs are labeled in the blank. The colors of the bars represent different clades. (C) MLST distribution in different toxin profiles. The height of the bars indicates the number of isolates. Each bar represents a single ST, except for the far-right bar containing all the singleton STs. The colors of the bars represent different toxin profiles.

nephrosis, and pneumonia. Significantly higher isolation rates were detected among patients with malignancy (26.3%, 36/137), IBD (25.7%, 53/206), and nephrosis (25.5%, 27/106), compared with those with pneumonia (5.2%, 5/97) (*P* < 0.05) (Fig. S2C and D).

**Phylogenetic structure of *C. difficile* isolates.** To better understand the characteristics of *C. difficile*, another 106 *C. difficile* isolates collected from the same hospital between December 1, 2015, and January 31, 2019 were included in the investigation, resulting in a total of 279 isolates. Overall, 270 (96.8%) isolates were successfully sequenced by next-generation sequencing and assigned to 36 STs, including 28 known and eight novel STs. Currently, the population structure of *C. difficile* comprises six major genomic clades (clades 1 to 5 and clade-CI) (15). The 28 known STs can be classified into four clades (clade 1, clade 4, clade 5, and clade-CI). To better analyze the phylogenetic structure of *C. difficile* isolates, we employed 13 reference genomes belonging to six clades. The pan-genome analysis identified 1529 core genes in ≥99% of the 283 genomes, less than a tenth of the total genes, indicating that the *C. difficile* genome has extreme levels of plasticity. A maximum likelihood phylogenetic tree was generated based on 20,057 cgSNPs, and 12 sequence clusters (SCs) were classified by hierBAPs analysis. The SCs ranged in size from 3 to 50 isolates and corresponded to the branches within the tree (Fig. 1A). Analysis of the phylogenetic tree positions and SCs revealed that the eight new STs were classified as clade 1 (ST812, ST813), clade 2 (ST820), and clade 4 (ST809, ST810, ST811, ST814). The results showed that ST3 isolates were separated into two distinct SCs, and eight SCs contained more than one ST. Among these, SC3 comprised multiple low-frequency STs (Fig. 1B).

**Toxic gene variants of *C. difficile*.** Based on virulence gene analysis, 64.8% (175/270) of isolates belonged to TCD, including the genotypes *tcdA*+*B*+*cdtA*−*B*− (146/

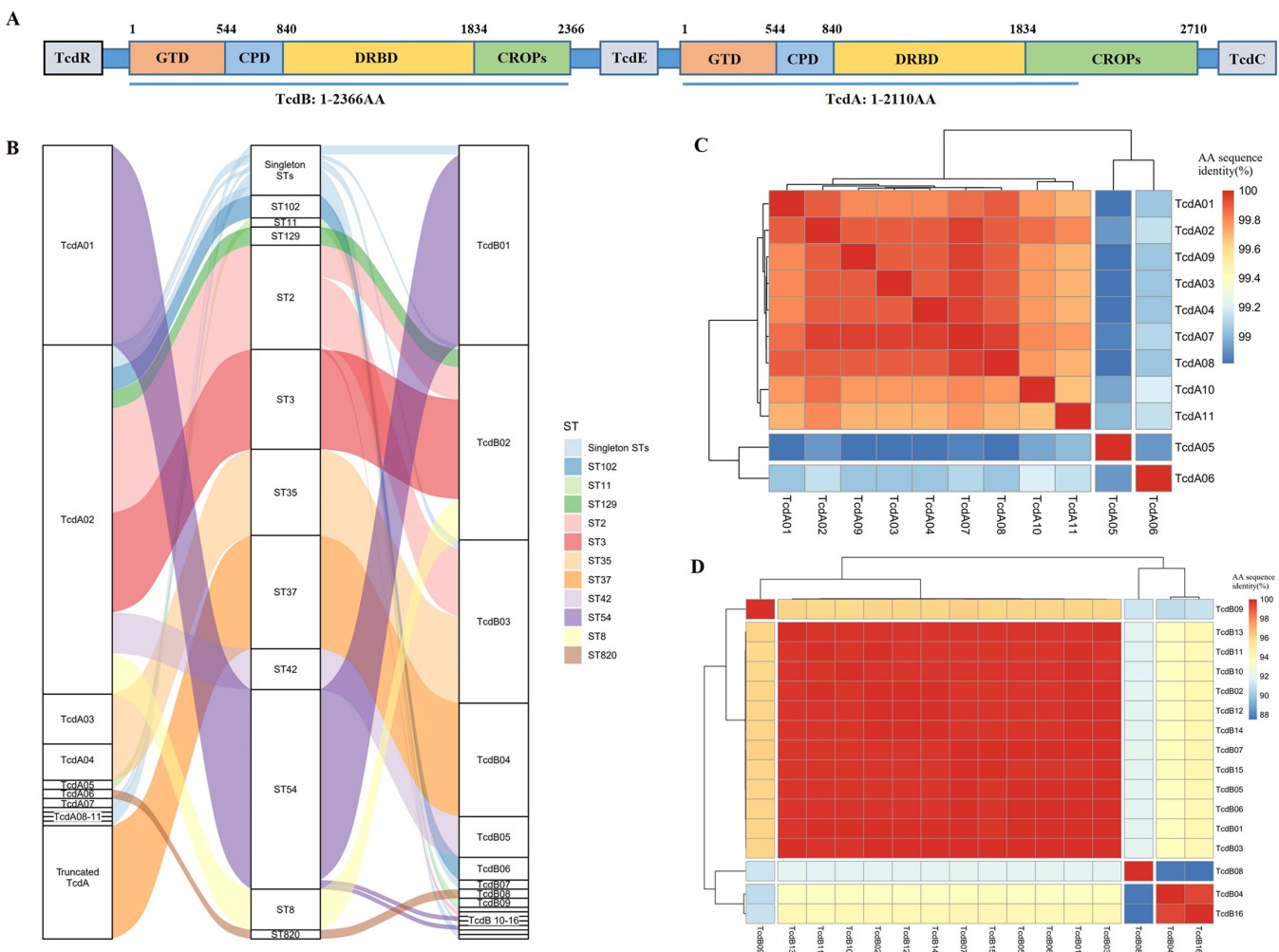

**FIG 2** The relationship between TcdA, TcdB variants, and MLST. (A) The genomic structure of PaLoc in the reference strain *C. difficile* CD630. (B) Riverplot graph showing the relationship between TcdA, TcdB variants, and MLST. The flows are colored according to MLST. (C) Amino acid sequence identity between the 11 types of TcdA variants. (D) Amino acid sequence identity between the 16 types of TcdB variants. Because the TcdA CROP domain contained many repeat sequences, the complete amino acid sequence was unable to be assembled, so only the first 6330 nucleotide sites (1 to 2110 aa) were included in the analysis.

175, 83.4%), *tcdA-B+cdtA−B−* (25/175, 14.3%), and *tcdA+B+cdtA+B+* (4/175, 2.3%). For TCD isolates, ST54 (44/175, 25.1%) was the most prevalent type, followed by ST37 (25/175, 14.3%), ST2 (23/175, 13.1%), and ST35 (19/175, 10.9%). ST39 (50/95, 52.6%) was the most prevalent ST among nontoxigenic *C. difficile* (NTCD), while ST3 was the only ST that contained both TCD (22/175, 12.6%) and NTCD (10/95, 10.5%) (Fig. 1C). Another toxin, namely, *C. difficile* transferase (CDT), was produced in so-called "hyper-virulent" strains (16). In total, four CDT-positive isolates were detected during this period, of which two isolates belonged to ST11 (clade 5) and two isolates belonged to a new sequencing type ST820 (clade 2).

The pathogenicity locus (PaLoc) is a 19.6 kb genetic locus that includes five toxin-associated genes: *tcdR*, *tcdB*, *tcdE*, *tcdA*, and *tcdC* (Fig. 2A). Genes *tcdA* and *tcdB*, encoding large *Clostridium* toxin TcdA and TcdB, are the major *C. difficile* virulence genes. Both TcdA and TcdB contain four functional domains: an N-terminal glucosyltransferase domain (GTD), cysteine protease domain (CPD), receptor-binding domain (DRBD), and C-terminal combined repetitive oligopeptides domain (CROPs) (17). In total, 11 types of TcdA variants and 16 types of TcdB variants were detected among the 175 TCD genomes. At the amino acid (AA) level, the minimal similarity of TcdA sequences was 98.8%, while it was only 87.5% for TcdB sequences, showing that the diversity of TcdB was higher than TcdA (Fig. 2C and D). The relationship between most STs and TcdA/B variants showed one-to-

one correspondence, except for ST35 (corresponding to TcdA03 and TcdA04), ST54 (corresponding to TcdB01, TcdB12, and TcdB14), and ST2 isolates (corresponding to TcdB02, TcdB03, and TcdB11) (Fig. 2B). However, there were only one or two AA substitutions between the variants that occurred in the same ST isolates, indicating strong consistency between STs and TcdA/B variants. In addition, six TcdR, three TcdE, and five TcdC variant types were identified. However, in certain STs, some genes were truncated on account of mutations. In ST37 isolates, all of TcdA and part of TcdR were truncated. In ST820 isolates, both TcdC and TcdE were truncated, while in ST11 isolates only incomplete TcdC was detected (Fig. S5).

**Antibiotic resistance spectrum of *C. difficile*.** Multidrug resistance is common among *C. difficile* isolates (10). To decipher the antibiotic resistance spectrum, we tested the antimicrobial susceptibility of the *C. difficile* isolates. Readouts were obtained for 96.1% (268/279) isolates. High resistance rates were shown for erythromycin (66.0%, 177/268), tedizolid (64.6%, 173/268), and clindamycin (62.3%, 167/268), followed by moxifloxacin (26.1%, 70/268), and rifaximin (9.7%, 26/268). Low resistance rates were shown for tigecycline (3.0%, 8/268), imipenem (0.8%, 2/268), and chloramphenicol (0.4%, 1/268). All isolates were susceptible to metronidazole, vancomycin, fidaxomicin, and amoxicillin-clavulanic acid (Fig. 3). In this study, 50.4% (135/268) of isolates were MDR, while only 10.1% (27/268) were susceptible to all the tested antibiotics. Interestingly, we found that the resistance rate was higher in NTCD than in TCD isolates for erythromycin, moxifloxacin, and rifaximin ($P < 0.05$); while the susceptibility for tedizolid was lower among TCD isolates than NTCD isolates ($P < 0.05$), and only TCD isolates were resistant to tigecycline. Although no isolates were resistant to vancomycin, the susceptibility was lower in TCD than in NTCD isolates ($P < 0.01$) (Table 1).

According to the ResFinder and CARD databases, a total of 26 antibiotic resistance genes (ARG) was detected, which conferred antibiotic resistance to aminoglycoside, chloramphenicol, lincosamide, trimethoprim, macrolide, streptothricin, and tetracycline. Erythromycin ribosomal methylase genes of class B (*ermB*), mediating resistance to antibiotics of the macrolide-lincosamide-streptogramin B (MLSB) family, were detected in 88% (154/175) of the erythromycin-resistant isolates and 90.9% (149/164) of the clindamycin-resistant isolates. Chloramphenicol O-acetyltransferase gene (*catP*) was detected in 11 isolates, which all showed lower susceptibility to chloramphenicol (MIC $\geq$ 8 mg/liter) (Fig. 3). DNA gyrase GyrA/GyrB and RNA polymerase RpoB confer mutational resistance to fluoroquinolones and rifamycin. All the 65 moxifloxacin-resistant strains had mutations in the *gyrA* or/and *gyrB* genes corresponding to T82I and D81N for GyrA and D426N and D426A for GyrB, and substitution on GyrB D426 only resulted in low-level resistance (MIC = 8 mg/liter). All the 26 rifaximin-resistant isolates had a mutation corresponding to a R505K substitution in RpoB, which was consistent with a previous report (Table 2) (12).

***C. difficile* nosocomial transmission.** Nosocomial transmission is one of the major routes of pathogen transmission. To elucidate the cause of the *C. difficile* epidemic, 251 isolates from inpatients from 22 distinct wards were included in the analysis (Fig. S3B). Up to 43.8% (110/251) of the isolates were genetically linked to the genetically closed previous case (single nucleotide variation [SNV] $\leq$ 3), while only 30.7% (77/251) of the isolates were genetically distinct from all the previous isolates, indicating *C. difficile* transmission could present in this hospital (Fig. 4A). A total of 335 genetically linked case pairs was identified (pairwise SNVs $\leq$ 3), while only nine (2.7%) case pairs were considered to have ward contact, 65 (19.4%) case pairs were likely caused by ward contamination, and 23 (6.9%) case pairs were considered to have hospital contact. The results were similar when we changed the threshold from SNV = 0 to SNV $\leq$ 10 (Fig. 4B). Based on SNVs $\leq$ 3, a total of 133 genetic clusters was classified, of which 30 clusters contained more than one isolate, and the largest cluster contained 15 isolates (Fig. 5A). The detailed in-hospital transmission routes were shown in the transmission network based on SNVs, ward, and sampling time (Fig. 5B). These results showed that there was no strong correlation between epidemiological linkages and genetic

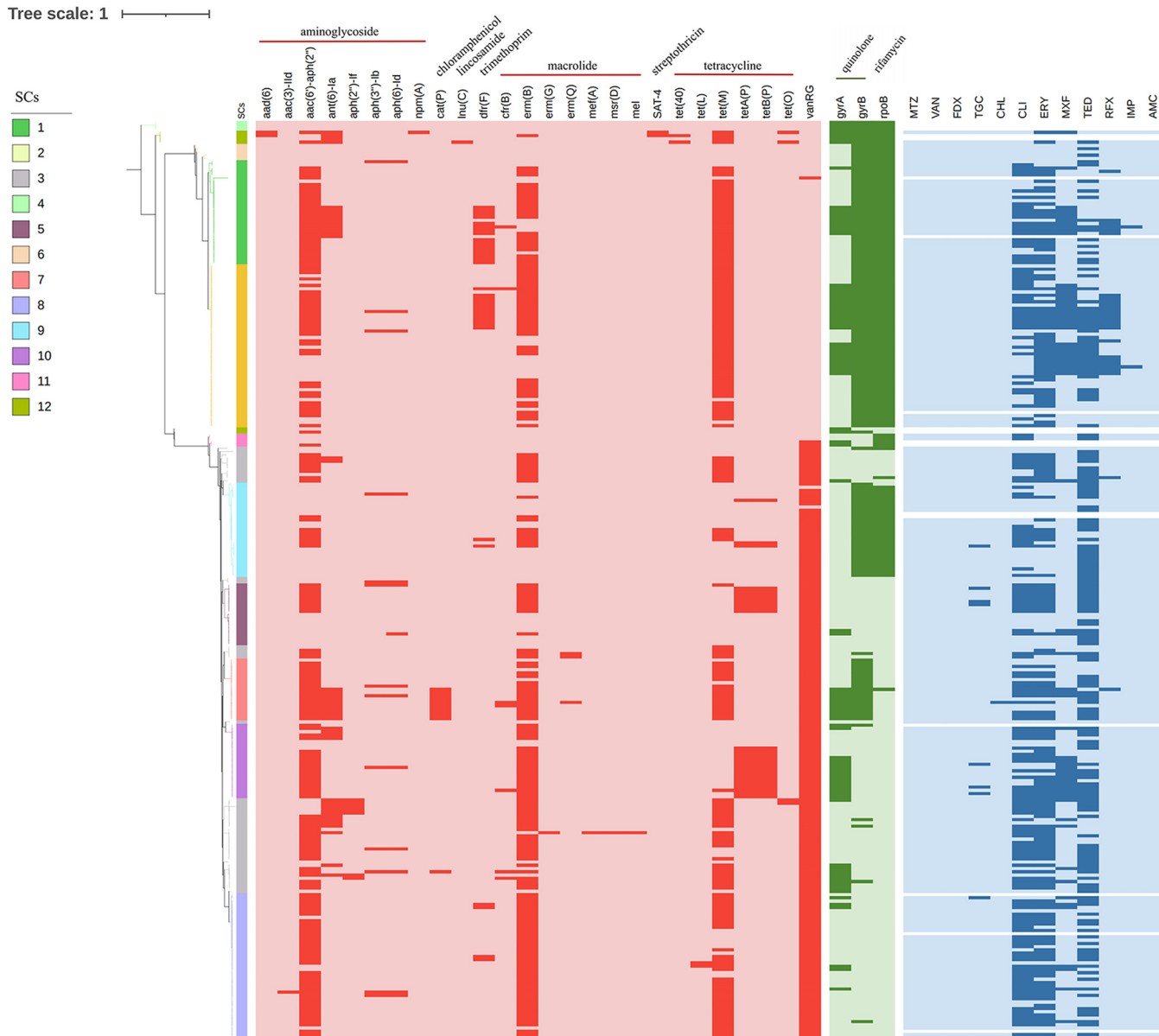

**FIG 3** Distribution of antimicrobial resistance elements. The phylogenetic tree was structured as in Fig. 1A. The branches and the color strip next to the tree are colored according to SCs. From left to right, the red heatmap represents the ARGs identified by the Resfinder and CARD databases, the green heatmap represents the amino acid substitutions, and the blue heatmap represents the resistant phenotype. The dark color indicates presence (resistance), and the light color indicates absence (susceptibility).

linkages, indicating the existence of other potential transmission routes, such as medical facility contamination or neglected intermediate hosts.

In total, 251 isolates were obtained from 234 inpatients, of which 14 inpatients had two or more isolates taken at different times. Patient 8, for whom a TCD isolate was detected, acquired an NTCD isolate after 297 days. For patient 2 and patient 4, NTCD isolates (pairwise SNVs ≤ 10) were detected for all tests, indicating a long period of colonization (270 and 288 days). For another 11 patients, TCD isolates were detected for all tests, indicating a colonization time ranging from 4 days to 326 days. Patient 1 and patient 9 appeared to be reinfected after 153 or 94 days, as the SNVs between the two TCD isolates were 45 and 24, respectively. The other nine patients were considered to have undergone relapse or had a long period of TCD colonization as the pairwise SNVs were < 10. These results showed that patients may be colonized with *C. difficile* for a longer period than previously expected, reaching at least 288 days with NTCD and 326 days with TCD isolates (Fig. 6).

**TABLE 1** Comparison of the susceptibilities of TCD and NTCD isolates to 12 antimicrobial drugs

| Isolates | Parameter | MTZ[a] | VAN | FDX | TGC | CHL | CLI | ERY | MXF | TED | RFX | IMP | AMC |
|---|---|---|---|---|---|---|---|---|---|---|---|---|---|
| | Resistance breakpoint (>) | 2 | 2 | 1 | 0.25 | 16 | 4 | 4 | 4 | 1 | 16 | 8 | 8 |
| TCD, 172 isolates | MIC range | ≤0.0625 to 1 | ≤0.0625 to 1 | ≤0.0625 to 0.5 | ≤0.0625 to 2 | ≤1 to 64 | ≤1 to >64 | ≤1 to >64 | ≤1 to 32 | ≤1 to >8 | ≤0.25 to >64 | ≤0.5 to >8 | ≤0.5 to 8 |
| | MIC$_{90}$ | 0.25 | 1 | 0.125 | 0.25 | 8 | >64 | >64 | 16 | 8 | ≤0.25 | 8 | 2 |
| | MIC$_{50}$ | 0.125 | 0.5 | ≤0.0625 | ≤0.0625 | 2 | 8 | 16 | ≤1 | 4 | ≤0.25 | 2 | ≤0.5 |
| | Resistance rate | 0.00% | 0.00% | 0.00% | 4.65% | 0.58% | 61.63% | 60.47% | 21.51% | 66.28% | 3.49% | 0.58% | 0.00% |
| NTCD, 96 isolates | MIC range | ≤0.0625 to 1 | ≤0.0625 to 1 | ≤0.0625 to 1 | ≤0.0625 to 0.25 | ≤1 to 16 | ≤1 to >64 | ≤1 to >64 | ≤1 to >64 | ≤1 to 8 | ≤0.25 to >64 | ≤0.5 to >8 | ≤0.5 to 8 |
| | MIC$_{90}$ | 0.25 | 0.5 | 0.125 | 0.125 | 4 | >64 | >64 | 16 | 8 | 64 | 8 | 2 |
| | MIC$_{50}$ | 0.125 | 0.25 | ≤0.0625 | ≤0.0625 | 2 | 16 | 64 | ≤1 | 2 | ≤0.25 | 4 | ≤0.5 |
| | Resistance rate | 0.00% | 0.00% | 0.00% | 0.00% | 0.00% | 63.54% | 76.04% | 34.38% | 61.46% | 20.83% | 1.04% | 0.00% |
| P value[b] | | 0.67 | **0.001** | 0.41 | **0.014** | 0.63 | 0.46 | **<0.001** | **0.049** | **0.029** | **<0.001** | 0.99 | 0.61 |
| P value[c] | | | | | | | 0.76 | **0.010** | **0.022** | 0.43 | **<0.001** | | |

[a]MIC (µg/ml); Metronidazole (MTZ), vancomycin (VAN), fidaxomicin (FDX), tigecycline (TGC), chloramphenicol (CHL), clindamycin (CLI), erythromycin (ERY), moxifloxacin (MXF), tedizolid (TED), rifaximin (RFX), imipenem (IMP), amoxicillin–clavulanate (AMC).

[b]Mann–Whitney test for MIC values.

[c]Chi-square test to analyze the resistance rate. P < 0.05 indicated in bold.

**TABLE 2** Amino acid substitutions in *gyrA*, *gyrB*, and *rpoB*

| Mutation sites | | Moxifloxacin (no. of isolates) | | | Mutation sites | Rifamycin (no. of isolates) | | |
|---|---|---|---|---|---|---|---|---|
| *gyrA* | *gyrB* | R[a] | S | ND | *rpoB* | R | S | ND |
| T82I | | 23 | | | H502N, R505K, I750M | 22 | | |
| T82I | S366A | 32 | | | R505K | 1 | | |
| T82I | S366A, R447K | 1 | | | R505K, I548M | 1 | | |
| T82I, M299V | | 1 | | | R505K, I750M | 2 | | |
| T82I, M324I | V130I | 3 | | | I750M | | 62 | 1 |
| T82I, K413N | Q160H, S366V, S416A | 1 | | | I750V | | 27 | 2 |
| D81N | | 1 | | | D1160E | | 2 | |
| D81N | S366A | 1 | | | 16 points[b] | | | 1 |
| | D426A | 1 | | | WT | | 148 | 1 |
| | D426N | 3 | | | | | | |
| M299V | D426N | 1 | | | | | | |
| | S366A | | 49 | 1 | | | | |
| | S366A, D465N | | 1 | | | | | |
| | I139R | | 28 | 2 | | | | |
| | V130I | | 9 | | | | | |
| M324I | V130I | | 7 | | | | | |
| M299V | | | 5 | | | | | |
| A99V, M299V | | | 1 | | | | | |
| K413N | Q160H, S366V, S416A | | 1 | | | | | |
| T581N | | | 1 | | | | | |
| 12 points[c] | 8 points[d] | | | 1 | | | | |
| WT | WT | | 95 | 1 | | | | |

[a]R, resistant; S, susceptible; ND, not detected.
[b]T227S, E291Q, D312E, A316D, D350N, S575A, E603D, N744S, D747E, Q748K, I750E, K751R, V951I, S1038T, E1019D, D1232E.
[c]N4K, V194I, L406Q, E410D, K413N, D444E, S478A, V546I, A613T, K628R, E664D, E693D.
[d]V130L, I139V, I348L, S366A, S416A, V563A, E581D, E586D.

As *C. difficile* can colonize the intestine of patients asymptomatically, the role of nosocomial transmission between CDI and asymptomatic colonized patients remains controversial (18–20). Here, we classified culture-positive cases with clinically significant diarrhea (≥3 times per 24 h or unshaped) as CDI and cases without diarrhea as *C. difficile* asymptomatic colonization (CDAC) (19). Of the 110 isolates genetically linked to a prior case, 50% of the isolates were associated with CDI cases, whereas 50% of the isolates were associated with CDAC cases (Fig. S6).

## DISCUSSION

According to the high mortality and relapse rates, an increasing number of studies have focused on CDI over the last 2 decades. The research has shown that *C. difficile* epidemiology and molecular types are diverse around the world (21). In China, most studies have only focused on the incidence and antibiotic-resistant phenotypes of *C. difficile* isolates. In this study, we performed a large genomic analysis of *C. difficile* isolates from patient samples taken over 4 years in a single health care center. Furthermore, detailed clinical information and epidemiological data were included to thoroughly investigate the strain characteristics and nosocomial transmission. To our knowledge, this is the first genomic study investigating the integrated characterization of *C. difficile* in a Chinese hospital.

Based on previous studies, young children have high rates of *C. difficile* carriage and advanced age is regarded as the risk factor for CDI (22, 23). Our results showed that the positive rate of detection of *C. difficile* in patients below 18 years of age was significantly higher than for other age groups, while the *C. difficile* carriage rate among the older patients was even lower than the median aged patients. It suggested that patients of any age had the potential for *C. difficile* carriage, which indicated the potential need for routine monitoring of this organism. Additionally, higher positive rates were detected in patients with malignancy, IBD, and nephrosis, which were associated with another CDI risk factor, hypo-immunity (22).

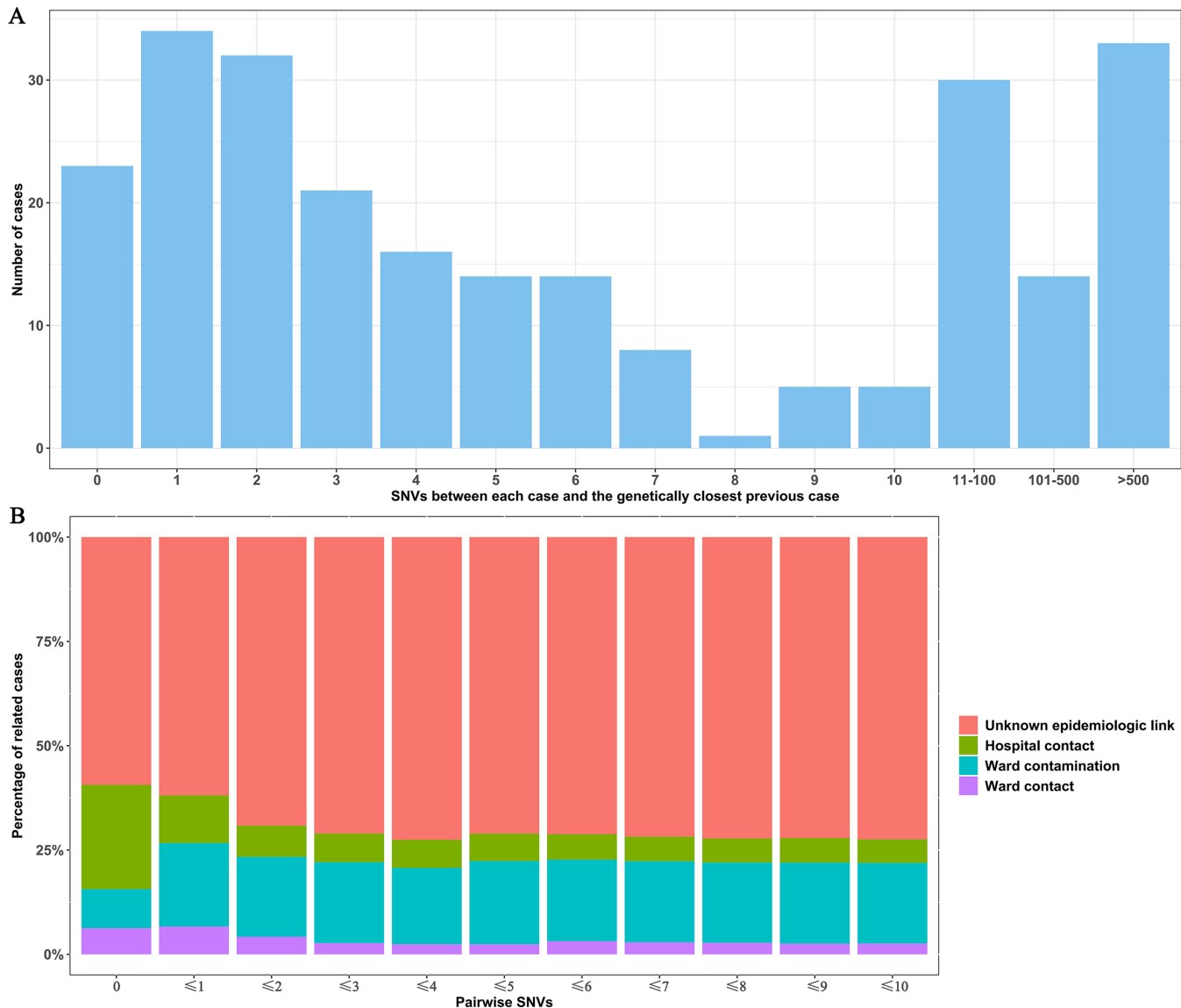

**FIG 4** The relationship between genetic linkages and epidemiological linkages. (A) SNVs between each case and the most genetically closed previous case. (B) The percentages of the epidemiological-linked case pairs in genetically linked case pairs according to the different SNV thresholds (from SNV = 0 to SNV ≤ 10). CD630 was used as the reference genome.

Global epidemics of CDI have shown that *C. difficile* is frequently transmitted around the world (24). However, molecular epidemiology is still diverse between different continents. Unlike the predominant type ST1 and ST11 in European and American countries, ST35, ST3, ST37, and ST54 were the most prevalent type in China (25, 26), which was generally consistent with our findings. In recent years, the hypervirulent isolate RT027, which is CDT-positive, has been increasingly reported (6). During our study period, two CDT-positive isolates, belonging to the new ST (ST820), were detected from a 55-year-old female patient during two separate admissions. The patient had severe diarrhea on both occasions but improved after vancomycin treatment. There were 20 and 191 aa substitutions detected in TcdA and TcdB, respectively, and *tcdC*, a reverse regulation gene that can inhibit toxin expression, was truncated in these isolates (Fig. S5). It is worth noting that these isolates are resistant to clindamycin, which is associated with the outbreak of *C. difficile* during the previous epidemic (27). Although no other ST820 isolates are detected subsequently, it is significant that a new hypervirulent strain has emerged in this region. A robust surveillance system is essential to prevent the outbreak of hypervirulent strains.

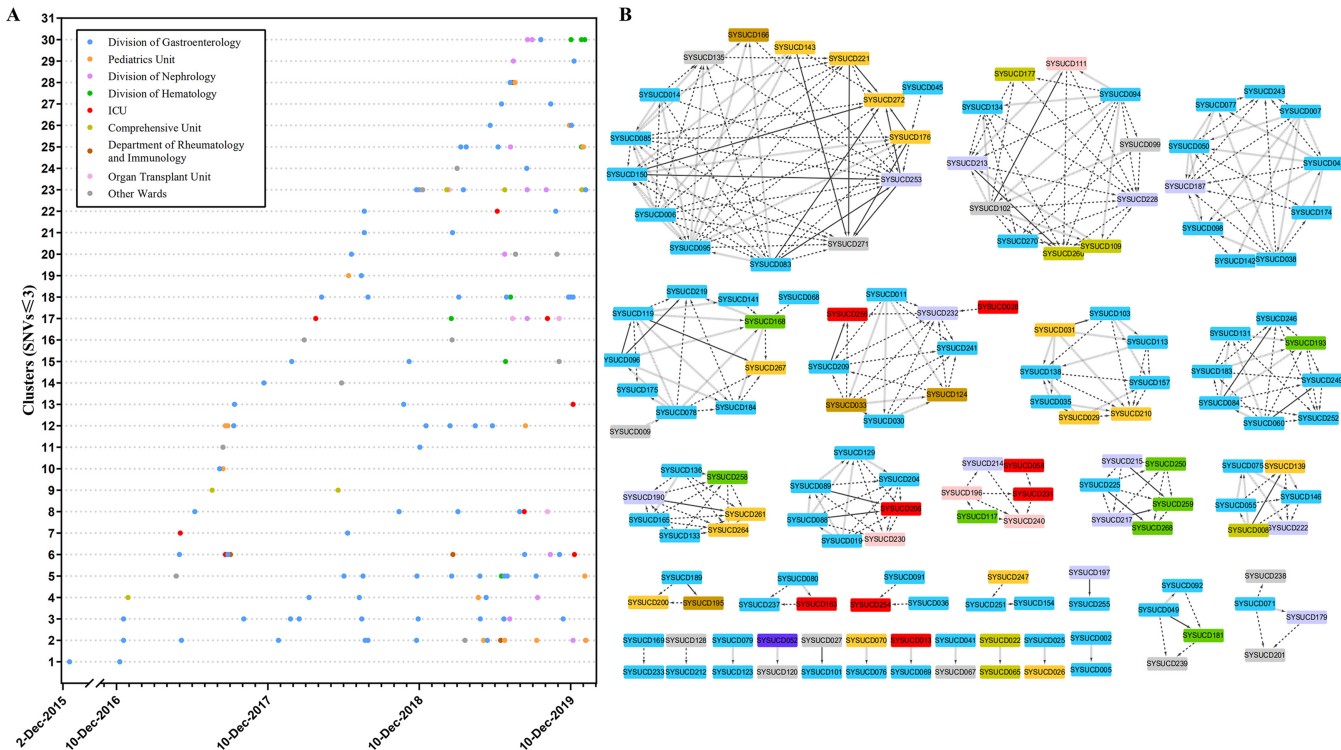

**FIG 5** Transmission networks of *C. difficile* positive inpatients based on SNVs, wards, and sampling time. Only 30 multiple case clusters are shown. Filled colors indicate the ward information. (A) The *x*-axis indicates the collection date. Each plot represents an isolate. Isolates sharing more than three SNVs with any prior isolates were defined as a distinct cluster and were plotted on a separate horizontal line. (B) The shape of arrows represents the sample interval: solid line (in 28 d), dashed line (28 d to 365 d), and vertical slash (>365). The arrowhead points to the isolates collected later.

In our study, 64.8% of the isolates were TCD and predominated by ST54, ST37, ST2, and ST35. Several studies suggested that *tcdA* and *tcdB* variants could affect the diagnosis, vaccine development, and even the disease severity (28–30). Multiple different toxin variants were detected among the 175 TCD isolates, containing 11 types of TcdA

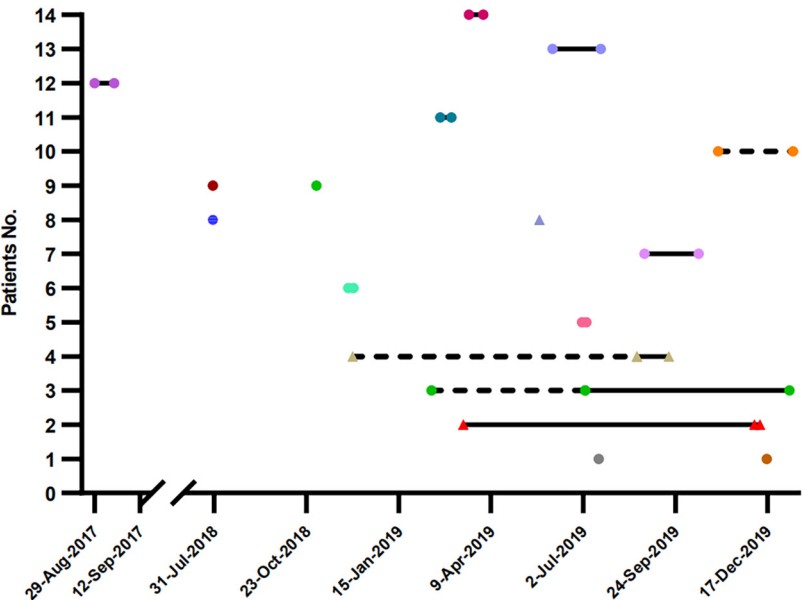

**FIG 6** Samples isolated from the same patient at different times. Each plot represents an isolate, and each line represents a patient. Genetically related isolates are shown in the same color. TCD isolates are represented by circles and NTCD isolates by triangles. Isolates connected with solid lines indicate 0 to 3 pairwise SNVs, while isolates connected with dashed lines indicate 4 to 10 pairwise SNVs.

and 16 types of TcdB variants, and the diversity of TcdB was higher than TcdA. A strong consistency was found between toxin variants and MLST, indicating that *tcdA* and *tcdB* were stable in *C. difficile* without frequent mutation and horizontal transmission. Meanwhile, a recent study considered that MLST, ribotype, and toxin variants were not always consistent (30), and another study suggested that the *tcdB* genes not only frequently mutate, but also continuously transfer and exchange among *C. difficile* strains (31). The toxin variants should be continuously monitored in future studies for a better understanding of the pathogenesis of *C. difficile* and the development of effective vaccines.

The PaLoc is replaced by a 115 bp noncoding region in NTCD isolates (32). To date, research has focused on NTCD isolates because they can acquire PaLoc via a conjugation-like mechanism (33). In our research, up to 35.2% of the isolates were NTCD, of which 52.6% belonged to ST39. The MIC results showed that the antibiotic resistance rates were higher in NTCD isolates for erythromycin, moxifloxacin, and rifaximin. Furthermore, up to 80.0% (24/30) of the moxifloxacin-resistant and 90% (18/20) of the rifaximin-resistant NTCD isolates belonged to ST39. Considering the high detection rate and antibiotic resistance rate of ST39 isolates, more attention should be paid to the molecular epidemiology and antibiotic resistance surveillance, in case of an outbreak of *C. difficile* ST39 in this region.

Because reference genome selection could influence the result of SNV calling (34), we used two reference genomes CD630 (clade 1) and M68 (clade 4) for calling the SNVs. The results were similar with each of the different reference genomes (Fig. 4 and Fig. S7). As clade 1 was predominant in our study, we decided to use CD630 as the reference genome for all SNV analyses. A high proportion (43.8%) of genetically linked isolates were detected in our study. We compared the pairwise SNVs from the same STs and the results showed that the most prevalent types in this study, ST39, and ST54, had the lower pairwise SNVs than other STs (Fig. S8). As the proportion of genetically linked isolates was associated with the local prevalent types (19), this could explain the larger fraction of genomic linkages in our study.

Patients with symptomatic infection or asymptomatic colonization may shed spores into the environment, where spores could persist for months (20, 35, 36). In this study, the proportions of isolates genetically linked to a prior CDI or CDAC case were equal, indicating CDI and CDAC patients had the same potential to cause the spread of *C. difficile*. In addition, our results showed that the colonization of *C. difficile* in patients could persist for almost a year, which was the potential risk of *C. difficile* in-hospital transmission. Therefore, routine screening of *C. difficile* on admission should be considered for all patients. Nonetheless, there were still many genetically related isolate pairs without evidence of contact, indicating the existence of other transmission routes. Further studies are required to better understand the transmission routes for developing more effective prevention strategies.

There are a few limitations to our study. First, only one colony was collected and sequenced from each sample; therefore, mixed infections by two or more *C. difficile* isolates were likely missed. Second, as *C. difficile* can form spores when in contact with oxygen, it can survive in a harsh environment such as the hospital environment for a long time. It may also colonize the intestinal tract of health care workers, relatives, and friends without symptoms. In this study, only patients were included for *C. difficile* screening, which could explain the lower proportion of epidemiological linkage among the genetically linked isolates. Further research should collect samples from the environment and close contacts, which might help to explain the nosocomial transmission more comprehensively.

In conclusion, on account of the high detection rates, serious antibiotic resistance, and diverse genome characteristics of the *C. difficile* isolates from the hospital in China, more attention should be paid to molecular epidemiological surveillance. As WGS is becoming a powerful tool for pathogen monitoring, a *C. difficile* surveillance system based on WGS should be established in China to prevent the nosocomial transmission of *C. difficile* and even outbreaks of infection.

## MATERIALS AND METHODS

**Strains and patients.** A total of 953 stool samples was collected between February 1, 2019 and January 31, 2020, from the First Affiliated Hospital of Sun Yat-sen University, a hospital with more than 2,000 beds in Guangzhou, China. After removing the duplicated samples, 173 *C. difficile* isolates were cultured successfully from 894 independent samples. To obtain a more comprehensive understanding of *C. difficile* characteristics in this region, another 106 isolates collected between December 1, 2015 and January 31, 2019 from the same hospital were included for further analysis, consisting of a total of 279 isolates. Detailed clinical information was collected after discharge.

**Isolation and identification.** Stool samples were cultured on taurocholate cycloserine-cefoxitin fructose agar in an atmosphere comprising 90% $N_2$, 5% $H_2$, and 5% $CO_2$ at 37°C for 48 h. The colonies were identified according to the typical morphology and odor of *C. difficile* and were confirmed by PCR of the housekeeping gene *tpi* and matrix-assisted laser desorption-ionization mass spectrometry. Multiplex PCR was performed to identify TCD isolates (37). Isolates that were either *tcdA* or *tcdB* positive were defined as TCD.

**Antimicrobial susceptibility testing.** MIC assays were performed by the agar dilution method as recommended by the Clinical and Laboratory Standard Institute (CLSI) with 12 antimicrobial agents. *C. difficile* ATCC 700057 was used as a control, at least in duplicate from independent cultures. The panels included metronidazole, vancomycin, fidaxomicin, tigecycline, rifaximin, clindamycin, erythromycin, chloramphenicol, moxifloxacin, tedizolid, imipenem, and amoxicillin-clavulanate. Susceptibility was interpreted based on the CLSI and European Committee on Antimicrobial Susceptibility Testing (EUCAST) breakpoints. Isolates resistant to three or more antibiotic classes were considered to be multidrug-resistant (MDR) according to the CLSI.

**Genome sequencing and analysis.** The *C. difficile* clinical isolates were grown in *Brucella* broth supplemented with vitamin K1 and chlorhematin for 18 to 24 h at 37°C anaerobically. DNA was extracted and purified using the Qiagen QiaAmp kit. Genomes were sequenced by Illumina technology. Raw reads were quality controlled by fastp and assembled by SPAdes (v.3.6.1) with the "–careful" parameter (38). Contigs ≥1 kb in length were kept. Kraken2 was used for decontamination, then Quast (v.4.4) was used for assessing the quality of assemblies, and Prokka (v.1.12) was used for annotation. STs were assigned by *C. difficile* PubMLST (https://pubmlst.org/cdifficile/), and new STs were submitted for assignment immediately. Pan-genome analysis was performed by Roary (v.3.12.0) (39). The core-genome single-nucleotide polymorphisms (cgSNPs) were extracted by SNP-sites (40). After strict filtration, a maximum likelihood phylogeny based on cgSNPs was constructed by RAxML (v.8.2.10) with 100 bootstrap replicates and visualized by iTOL (https://itol.embl.de/). Hierarchical bayesian analysis of population structure (hierBAPS) was carried out to assign sequence clusters (SCs) based on the core genome.

**Virulence gene variants analysis.** The virulence genes were detected by ABRicate (v.0.8.7) based on the Virulence Factor Database (VFDB). The *tcdA* (1 to 6330 bp), *tcdB* (7098 bp), *tcdC* (699 bp), *tcdR* (185 bp), and *tcdE* (501 bp) gene sequences were obtained by BLASTn, which searched *de novo* assemblies compared with reference sequences from *C. difficile* strain CD630 (GenBank accession no. AM180355.1). Then, the sequences were translated into amino acid (aa) sequences for variant classification. The R package "ggalluvial" and "pheatmap" were applied to draw the river plots and heatmaps.

**Resistance gene analysis.** The resistance genes were detected by ABRicate (v.0.8.7) based on the Resfinder and CARD databases. For detecting mutations, the *gyrA*, *gyrB*, and *rpoB* sequences were extracted and compared with the reference sequences from *C. difficile* strain CD630 with BLASTn. Only nonsynonymous mutations were kept for further analysis. The heatmap was visualized by iTOL.

**Calling of single-nucleotide variants.** For calling the single-nucleotide variants (SNVs), the raw reads were mapped against the reference genome sequence of *C. difficile* strain CD630 and M68 (GenBank accession no. FN668375.1) through Burrows-Wheeler alignment (BWA) (41). After sorting the reads by SAMtools, Gatk HaplotypeCaller was employed to call the SNVs. Gatk VariantFiltration was then performed to obtain high-quality SNVs, using the parameters "-filter 'QUAL < 30.0' -filter 'QD < 2.0' -filter 'MQ < 40.0' –window 10 –cluster 2."

**Transmission analysis.** Based on the analyses by Eyre et al. (42), we considered that isolates with 0 to 3 SNVs were genetically linked. Isolates were considered genetically distinct if they possessed more than 10 SNVs. Patients who shared time (two positive samples collected within 28 days) on the same ward were considered having ward contact. Patients admitted to the same ward, but up to 28 days apart, were considered to have possible ward contamination. Ward contamination was assumed to persist for 365 days. Patients who shared time in the same hospital without ward contact were considered having hospital contact (43). For patients from whom multiple samples were taken, more than 10 SNVs between isolates were considered a new strain (42). Boxplot and bar charts were created by R package "ggplot2," scatterplots were created by GraphPad Prism 8 software and transmission networks were performed by Cytoscape (v.3.8.2).

**Statistical analysis.** Statistical analysis was performed using GraphPad Prism 8 software by a Mann–Whitney U test and chi-square test ($n ≥ 40$; with Yale's correction for continuity if $1 < T ≤ 5$). $P < 0.05$ was considered statistically significant.

**Data availability.** Sequences generated during this study can be found in the NCBI database under BioProject PRJNA686004.

## SUPPLEMENTAL MATERIAL

Supplemental material is available online only.

**SUPPLEMENTAL FILE 1**, PDF file, 2.5 MB.

## ACKNOWLEDGMENTS

We thank International Science Editing (http://www.internationalscienceediting.com) for editing the manuscript.

This work was supported by the National Natural Science Foundation of China (grant numbers 82061128001 and 81830103), National Key Research and Development Program (grant number 2017ZX10302301), Guangdong Natural Science Foundation (grant number 2017A030306012), Project of high-level health teams of Zhuhai at 2018 (The Innovation Team for Antimicrobial Resistance and Clinical Infection), 111 Project (grant number B12003), and Open Project of Key Laboratory of Tropical Disease Control (Sun Yat-sen University), Ministry of Education (grant numbers 2020kfkt04 and 2020kfkt07).

Ethical approval was sought and given by Sun Yat-Sen University ZhongShan School of Medicine (November 1, 2014).

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
