## [Reviewer comments · Microbiology Spectrum]

Microbiology Spectrum

Whole-genome sequencing reveals the high nosocomial transmission and antimicrobial resistance of *Clostridioides difficile* in a single centre in China, a four-year retrospective study

Xin Wen, Cong Shen, Jinyu Xia, Lan-lan Zhong, Zhongwen Wu, Mohamed El-Sayed Ahmed, Nana Long, Furong Ma, Guili Zhang, Wenwei Wu, Jianlve Luo, Yong Xia, min dai, Liyan Zhang, Kang Liao, Siyuan Feng, Cha CHEN, Wenji Luo, Yishen Chen, and Guo-Bao Tian

Corresponding Author(s): Guo-Bao Tian, Sun Yat-sen University

Review Timeline:

Submission Date:	August 19, 2021
Editorial Decision:	October 6, 2021
Revision Received:	November 7, 2021
Editorial Decision:	November 12, 2021
Revision Received:	November 21, 2021
Accepted:	November 23, 2021

Editor: S. Wesley Long

Reviewer(s): The reviewers have opted to remain anonymous.

Transaction Report:

DOI: <https://doi.org/10.1128/spectrum.01322-21>

October 6, 2021

Prof. Guo-Bao Tian
Sun Yat-sen University
Zhongshan School of Medicine
Zhongshan School of Medicine, Sun Yat-sen University
74 Zhongshan 2nd Road
Guangzhou, China 510080
China

Re: Spectrum01322-21 (Whole-genome sequencing reveals high nosocomial transmission and serious antimicrobial resistance of *Clostridioides difficile* in a single centre in China, a four-year retrospective study)

Dear Prof. Guo-Bao Tian:

Thank you for submitting your manuscript to Microbiology Spectrum. When submitting the revised version of your paper, please provide (1) point-by-point responses to the issues raised by the reviewers as file type "Response to Reviewers," not in your cover letter, and (2) a PDF file that indicates the changes from the original submission (by highlighting or underlining the changes) as file type "Marked Up Manuscript - For Review Only". Please use this link to submit your revised manuscript - we strongly recommend that you submit your paper within the next 60 days or reach out to me. Detailed information on submitting your revised paper are below.

Link Not Available

Sincerely,

S. Wesley Long

Journals Department
Reviewer comments:

Reviewer #1 comments are supplied in an attachment.

Reviewer #2 (Public repository details (Required)):

Whole genome sequences should be deposited.

Reviewer #2 (Comments for the Author):

Wen and colleagues present a genomic epidemiology analysis of *C. difficile* colonization in a large medical center in China. A total of 953 stool samples were screened over the 1-year study period, with *C. difficile* cultured from 173. In addition, 106 isolates from the prior 4 years were included. Genomic analysis revealed the following key results: 1) the diversity of sequence types is quite different than what has been described in the U.S. and Europe, with a dominance of clade 4, 2) the colonization rate was high - with ~20% of patients harboring *C. difficile*, 3) both toxin and non-toxin producing strains were identified, and had differences in their antibiotic resistance profiles, 4) extensive genomic linkages were identified between patients, suggesting

significant transmission was occurring and 5) almost 50% of colonized individuals were deemed to have developed *C. difficile* infection (CDI).

Overall, this is a very nice study, with an impressive sample size, solid analysis and intriguing results. My comments are largely to provide more details regarding the study design and add additional analyses to expand on and/or provide context to some of the more provocative findings.

Major comments

1. More details should be provided on the sample collection strategy. In particular: 1) how were patients identified for screening? 2) what fraction of patients were screened? and was there variability in the comprehensiveness of screening among wards or over time?
2. Overall, the author's find a significantly larger fraction of genomic linkages than has been reported in comparable studies of *C. difficile* colonization and/or infection. While this could be due to differences in local strain types or epidemiology, some additional analyses to confirm robustness of these findings are warranted. Studies in both Europe (Eyre et al., *Clinical Infectious Diseases*, 2018) and the U.S. (Miles-Jay et al., *Microbial Genomics*, 2021) have found that variant distances can be misleading for certain strain types, as identical strains can be observed across large temporal or geographic distances. To provide additional context to the detected genetic linkages, I would advise incorporating in the analysis publicly available genomes from high-prevalence STs, preferable ones that have been sequenced in China. In addition, plots showing how genetic distance relates to the time between isolates would also provide a sense of how likely direct or indirect transmission is to account for genetic linkages among hospitalized patients.
3. To provide context for the patient pairs deemed to be linked by transmission, it would be helpful to show histograms of genetic distances among isolate pairs from the same sequence type. This will give a sense of how much standing diversity there was.
4. The author's report a shockingly high rate of infection among colonized cases (45%). As this is more than an order of magnitude larger than previous estimates, more details should be reported. In particular: 1) how was the definition of clinical CDI different than prior reports of infection incidence among *C. difficile* carriers? 2) were *C. difficile* infections restricted to toxigenic strains, or did they also include non-tox? 3) were there any clinical variables associated with infection? 4) were any STs associated with infection? Also, added discussion of possible sources of this finding would be valuable.

Minor comments

1. In Table 1 the p-values should be adjusted to a standard number of significant figures

Staff Comments:

Preparing Revision Guidelines

Please return the manuscript within 60 days; if you cannot complete the modification within this time period, please contact me. If you do not wish to modify the manuscript and prefer to submit it to another journal, please notify me of your decision immediately so that the manuscript may be formally withdrawn from consideration by Microbiology Spectrum.

Corresponding authors may join or renew ASM membership to obtain discounts on publication fees. Need to upgrade your

membership level? Please contact Customer Service at Service@asmusa.org.

In this manuscript, Wen et al. present a genomic epidemiological picture of *C. difficile* carriage, infection, and transmission in a large hospital in Guangzhou, China. The authors describe the prevalence, strain diversity, AMR and virulence types, and evidence of transmission of *C. difficile* in their hospital. This manuscript addresses a paucity of data regarding the molecular epidemiology of *C. difficile* in China, which is a valuable contribution. However, several of the conclusions drawn from these data are too strong or not valid, as indicated below. Additionally, more information is needed in order to properly interpret the transmission inference analyses.

Major Comments:

- The observation in line 184 about “the isolation rates of *C. difficile* showed a declining trend with age” and later interpretation that this opposed previous reports (line 188 and line 322) is not valid. It is well documented that young children, in particular, have high rates of *C. difficile* carriage (both toxin-producing and non-). The proportion of samples screened that returned *C. difficile* also do not directly speak to the “susceptibility” to CDI. The overall young median age of those with *C. difficile* collected is intriguing, but it isn’t meaningful without also knowing the age distribution among all screened patients. The manuscript would benefit from a Table 1 that described some basic clinical and demographic characteristics of the screened patients.
- The analyses regarding the relationship between genetic linkages and epidemiological linkages (time spent on the same ward) require clarification. What proportion of genetically linked patients had ward-linkages, compared to patients generally? How was the result on lines 279-281 determined? Also the described genetic linkages do not correspond directly to “transmission rates” as stated in line 343 or “high rates of nosocomial transmission” as described in lines 70-71.
- Were linked clusters restricted to particular strain types? Particular wards? Were most of them 0, 1, 2, or 3 SNVs? Is there more overlap between ward contact and genomic linkage if the authors decrease their SNV threshold to 1 or 2 SNVs? It would be helpful to break down some of this information, at least in the supplementary materials, in order to help distinguish between different epidemiological phenomena that are underlying the high rate of genomic linkages, as well as to understand how much the conclusions might change with a slightly more strict SNV threshold.
- What is the ward distribution of those screened? Figure S3 can’t be interpreted without this information.
- As the authors note, *C. difficile* is a very diverse species and it is known that a distantly related reference genome can lead to false negative variant calls (<https://academic.oup.com/gigascience/article/9/2/giaa007/5728470>). The authors should consider repeating the analyses with more reference genomes (such as one reference genome per clade). If not, they should acknowledge that their estimates of linkages could be inflated due to this limitation.

- Line 357 – detection of one previously undefined strain with CDT toxin is not equivalent to the “emergence of a hypervirulent strain”. Many other factors should be considered when considering hypervirulence, including clinical manifestations on a population-level. Please remove this reference, as it is an overinterpretation of the data.

Minor Comments:

- Lines 66-67 – what does “a serious antibiotic resistance condition” mean?
- What is the significance of the description of the different toxin gene types and how they relate to ST (Lines 225-241 and Figure 2)? These data are not interpreted or raised in the discussion, so I don’t know what to make of them.
- It is stated that all isolates were sensitive to vancomycin (Line 250) but that NTCD isolates were more frequently resistant to vancomycin (Line 253). How these both true? What is being compared to make this determination? This doesn’t appear to be stated in Table 1.
- There is no discussion about the clinical implications of the antimicrobial resistance genes detected and described—while antimicrobial resistance is always concerning, it would be helpful to know how the described resistance patterns relate to the standard treatment of *C. difficile* in the described setting.
- Line 338-339—why does ST39 have “epidemic potential”? I value the call for molecular surveillance, but stating it “may present a serious threat to human health” based off of the presented data is too strong.
- Line 343-345 doesn’t make sense—“which was the hidden danger of hospital transmissions”? Consider re-phrasing.
- Lines 353-356 – how do these stated limitations affect the interpretations in this study?

Cover Letter

Editor

Microbiology Spectrum

Subject: Spectrum01322-21

Dear Editor S. Wesley Long,

Thank you for considering our manuscript: **“Whole-genome sequencing reveals high nosocomial transmission and serious antimicrobial resistance of *Clostridioides difficile* in a single centre in China, a four-year retrospective study”** for publication in **Microbiology Spectrum**.

We have addressed the specific concerns of the reviewers, incorporated their suggestions, and uploaded some new files: (1) A file named “Response to Reviewer Comments” contained detailed point-for-point response to the comments, (2) A file of the revised manuscript, (3) A file named "Marked Up Manuscript File” contained all the marked changes, (4) Figure files in TIFF format, (5) A file named “Description of Supplementary Files” contained the supplementary files and the description of them, (6) A file named “cover letter” which has a little changed based on the comments.

We believe that this study significantly adds to our understanding of *C. difficile* genomic epidemiology in China and warrants a quick publication in your prestigious journal.

Please do not hesitate to contact us if additional information is required. Thank you for consideration.

Sincerely yours,

Guo-Bao Tian Ph.D., Professor

Email: tiangb@mail.sysu.edu.cn; guobaotian@gmail.com

Zhongshan School of Medicine

Sun Yat-sen University

74 Zhongshan 2nd Road

Guangzhou 510080, China

Response letter

#REVIEWERS' COMMENTS

We thank the reviewers for the constructive comments that are helpful to improve our manuscript. The point-for-point responses to the comments are given below (The comments in this document are bold and italic).

Reviewer #1

In this manuscript, Wen et al. present a genomic epidemiological picture of C. difficile carriage, infection, and transmission in a large hospital in Guangzhou, China. The authors describe the prevalence, strain diversity, AMR and virulence types, and evidence of transmission of C. difficile in their hospital. This manuscript addresses a paucity of data regarding the molecular epidemiology of C. difficile in China, which is a valuable contribution. However, several of the conclusions drawn from these data are too strong or not valid, as indicated below. Additionally, more information is needed in order to properly interpret the transmission inference analyses.

Thank you for the carefully reading of our manuscript and raising the constructive suggestions. We have replied the comments point-for-point and revised our manuscript following the comments to make it more valid and comprehensive.

Major Comments:

1. The observation in line 184 about “the isolation rates of C. difficile showed a declining trend with age” and later interpretation that this opposed previous reports (line 188 and line 322) is not valid. It is well documented that young children, in particular, have high rates of C. difficile carriage (both

toxin-producing and non-). The proportion of samples screened that returned C. difficile also do not directly speak to the “susceptibility” to CDI. The overall young median age of those with C. difficile collected is intriguing, but it isn’t meaningful without also knowing the age distribution among all screened patients. The manuscript would benefit from a Table 1 that described some basic clinical and demographic characteristics of the screened patients.

Reply: Thank you for your valuable suggestions to improve the quality of our manuscript. We admitted that the young children had high rates of *C. difficile* carriage which corresponded to our result, and we agreed with the reviewer’s words “*The proportion of samples screened that returned C. difficile do not directly speak to the ‘susceptibility’ to CDI*”. Based on the reviewer’s comment, we deleted the sentence “Surprisingly, this was the opposite trend to previous reports that older patients were more susceptible to CDI” in previous lines 187-188, and added a new Table S1 which described the characteristics of all the screened patients included the age distribution among all screened patients, CD positive patients, TCD positive patients and CD negative patients. And revised the discussion part in lines 312-316.

2. The analyses regarding the relationship between genetic linkages and epidemiological linkages (time spent on the same ward) require clarification. What proportion of genetically linked patients had ward-linkages, compared to patients generally? How was the result on lines 279-281 determined? Also the described genetic linkages do not correspond directly to “transmission rates” as stated in line 343 or “high rates of nosocomial transmission” as described in lines 70-71.

Reply: Thank you for this insightful comment. We appreciated that the reviewer suggested us to calculate the proportion of genetically linked patients had ward-linkages compared to patients generally. Among the genetically linked cases, 2.7% case pairs were considered to have ward contact, 19.4% case

pairs could be affected by ward contamination, and 6.9% case pairs were considered to have hospital contact (lines 276-279). While for all the inpatients cases, 2.0% case pairs were considered to have ward contact, 16.4% case pairs were likely affected by ward contamination, and 5.4% case pairs were considered to have hospital contact (data not shown in the manuscript). The proportion of ward-linkages was higher among the genetically linked patients than patients generally, indicating the possibility of nosocomial transmission. The result in previous lines 279-281 determined by calculating the proportion of epidemiological linkages among genetically linked cases, and we revised this part in lines 276-280 and built a new Figure 4B for better exhibiting this result. We agreed the reviewer's opinion that ***“the genetic linkages do not correspond directly to transmission rates”***. To better describe this result, we replaced ***“high transmission rates were shown for both CDI and asymptomatic colonization patients”*** (previous line 343) by “the proportions of isolates genetically-linked to a prior CDI or CDAC case were equal, indicating CDI and CDAC patients had the same potential to cause *C. difficile* spreading” (lines 364-366), and changed ***“a high nosocomial transmission”*** (previous lines 70-71) to “the unknown nosocomial transmission risk” (lines 71-72) based on the lower proportion of epidemiological linkages among genetically linked cases.

3. Were linked clusters restricted to particular strain types? Particular wards? Were most of them 0, 1, 2, or 3 SNVs? Is there more overlap between ward contact and genomic linkage if the authors decrease their SNV threshold to 1 or 2 SNVs? It would be helpful to break down some of this information, at least in the supplementary materials, in order to help distinguish between different epidemiological phenomena that are underlying the high rate of genomic linkages, as well as to understand how much the conclusions might change with a slightly more strict SNV threshold.

Reply: Thank you for the reviewer's comment. To better explain the relationship between the genetic

linkage and epidemiological linkage, we added the new Figure 4, Figure 5B and Figure S8. The new Figure 4 answered the question that “*Were most of them 0, 1, 2, or 3 SNVs? Is there more overlap between ward contact and genomic linkage if the authors decrease their SNV threshold to 1 or 2 SNVs?*”. The figure showed the frequency distribution of SNVs between each case and the genetically-closest previous case (panel A), and the relationship between different genetic linkages (from SNV=0 to SNV≤10) and epidemiological linkages (panel B). When the threshold was SNV≤3, the rate of the epidemiological linkage among the genetically-linked case pairs became stable. Meanwhile, the study (Eyre *et al.*, *N Engl J Med*, 2011) determined that 0 to 2 or 3 SNVs would be expected between genetically-linked isolates by estimating the evolutionary rate of *C. difficile*. So we chose SNV≤3 to determine the genetically-linked cases. The new Figure 5B answered the question that “*Were linked clusters restricted to particular wards?*”. The results showed that linked clusters were not restricted to particular wards. The new Figure S8 answered the question that “*Were linked clusters restricted to particular strain types?*”. The result showed that in the high-prevalence STs, the pairwise SNVs were lower than the other STs, which could explain the high proportion of genetically-linked isolates in this study.

4. What is the ward distribution of those screened? Figure S3 can't be interpreted without this information.

Reply: Thank you for making this valuable suggestion. First, I have to clarify the isolates exhibited in the previous Figure S3. They included two parts of isolates, which were 173 isolates separated from the 894 screened cases and 106 isolates collected between December 2016 and January 2019 from the same hospital (lines 105-118). Excluding the sequencing failed and the outpatient cases, a total of 251 isolates were included in the analysis of nosocomial transmission (lines 271-273). For better exhibiting the

patients' characteristics, we revised Figure S3 into two panels: panel A showed the wards distribution of the 857 fecal samples collecting from inpatient cases (24 distinct wards), and panel B showed the wards distribution of 251 isolates separated from inpatients (22 distinct wards).

5. As the authors note, C. difficile is a very diverse species and it is known that a distantly related reference genome can lead to false negative variant calls (<https://academic.oup.com/gigascience/article/9/2/giaa007/5728470>). The authors should consider repeating the analyses with more reference genomes (such as one reference genome per clade). If not, they should acknowledge that their estimates of linkages could be inflated due to this limitation.

Reply: Thanks for the reviewer's suggestion. As the link (<https://academic.oup.com/gigascience/article/9/2/giaa007/5728470>) reported, the reference genome selection had an influence on SNVs calling. In our study, most of the isolates were belonged to clade1 (66.7%, 180/270), following by clade4 (31.5%, 85/270). So we chose the *Clostridium difficile* CD630 (clade1) as the reference genome for SNVs calling. Following the reviewer's suggestion, we chose another reference genome *Clostridium difficile* M68 (clade4) to call the SNVs for detecting the influence of reference genome on the SNVs value. The result was almost the same when using M68 as the reference genome, and the added result was shown in lines 354-358 and the new Figure S7. We didn't choose reference genome from the other four clades as isolates of these clade were few in our study.

6. Line 357 – detection of one previously undefined strain with CDT toxin is not equivalent to the “emergence of a hypervirulent strain”. Many other factors should be considered when considering hypervirulence, including clinical manifestations on a population-level. Please remove this reference, as it is an overinterpretation of the data.

Reply: Thanks for your comments. We had deleted the words “*emergence of a hypervirulent strain*” in the previous line 357, and rewritten the paragraph as “In conclusion, on account of the high detection rates, serious antibiotic resistance and diversity genome characteristics of the *C. difficile* in hospital in China, more attention should be paid on the molecular epidemiology surveillance. As WGS is becoming a powerful tool for pathogen monitoring, a *C. difficile* surveillance system based on WGS should be established in China to prevent the nosocomial transmission and even *C. difficile* outbreaks” (lines 382-386).

Minor Comments:

1. Lines 66-67 – what does “a serious antibiotic re-sistance condition” mean?

Reply: We have changed the “*a serious antibiotic resistance condition*” to “known to be resistant to multiple antibiotics” (lines 66-67).

2. What is the significance of the description of the different toxin gene types and how they relate to ST (Lines 225-241 and Figure 2)? These data are not interpreted or raised in the discussion, so I don't know what to make of them.

Reply: *tcdA* and *tcdB* are the most significant toxin factors in the pathogenic process of CDI. Several studies suggested that *tcdA* and *tcdB* variants could affect the diagnostic and vaccine development (*Mansfield et al., PLoS Pathog, 2020*), and *tcdB* variants could influence the disease severity (*Lanis et al., PLoS Pathog, 2013*). So it is important to clarify the *C. difficile* toxin gene variants for better diagnosis and treatment of CDI. In our study, multiple different toxin variants were detected, and the diversity of TcdB was higher than TcdA. A strong consistence between toxin variants and MLST has been described in lines 231-236. However, a recent study (*Zhenghui Li et al., Frontiers in Microbiology, 2020*) considered that MLST, ribotype and toxin variants were not always consistent, researchers should combine them together to perform the diagnosis and vaccine development. Another

study suggested that the *tcdB* genes not only frequently mutate, but also continuously transfer and exchange among *C. difficile* strains (Shen E et al., *Communications Biology*, 2020). The toxin variants should be continuously monitoring in the future studies for better understand the pathogenesis of *C. difficile* and developing the effective vaccines. On account of this result is not the most vital part in our research, we didn't discuss it in-depth. While, thanks for the reviewer's suggestion, we have added the discussion part on the significance of description the toxin gene types and how they related to MLST (lines 334-344).

3. It is stated that all isolates were sensitive to vancomycin (Line 250) but that NTCD isolates were more frequently resistant to vancomycin (Line 253). How these both true? What is being compared to make this determination? This doesn't appear to be stated in Table 1.

Reply: Thank you for this comment, we have corrected the expression to “Though no isolates was resistant to vancomycin, the susceptibility was lower for TCD than NTCD isolates ($p < 0.01$)” (lines 253-254) and in the abstract “Though none vancomycin resistant isolates was detected, the MIC-value to vancomycin was higher for toxigenic isolates ($p < 0.01$)” (lines 57-58). In our study, all the isolates are susceptible to vancomycin according to the CLI standard as $MIC \leq 2$. While when we compared the MIC value between TCD and NTCD isolates by Mann-Whitney U test, we found that the MIC value to vancomycin was significantly higher for TCD than NTCD isolates, indicating the susceptibility was lower for TCD isolates. The statistical result was shown in the last two rows of Table 1, where we used chi-square test for comparing the resistance rates, and used Mann-Whitney U test for comparing the MIC-values.

4. There is no discussion about the clinical implications of the antimicrobial resistance genes detected

and described—while antimicrobial resistance is always concerning, it would be helpful to know how the described resistance patterns relate to the standard treatment of C. difficile in the described setting.

Reply: Metronidazole and vancomycin are the first-line antibiotic for CDI, and nowadays fidaxomicin is also used on treating CDI. All isolates in our study were susceptible to metronidazole, vancomycin and fidaxomicin, and none of the antimicrobial resistance genes to these antibiotics was detected. For the other antibiotics like erythromycin and clindamycin, chloramphenicol, moxifloxacin, and rifaximin, corresponding antimicrobial resistance genes *ermB*, *catP*, *gyrA/B* and *rpoB* had been described in the result parts (lines 257-268).

5. Line 338-339—why does ST39 have “epidemic potential”? I value the call for molecular surveillance, but stating it “may present a serious threat to human health” based off of the presented data is too strong.

Reply: Thank you for the reviewer’s suggestion. Antibiotic resistance was associated with the outbreak of *C. difficile*, for the most significant example as the emergence of fluoroquinolone-resistant lineages of *C. difficile* RT027 causing the outbreaks in the northern hemisphere in the early 2000s (*McDonald et al., N Engl J Med, 2005*). As the detection rate and antibiotic resistance rates were high for ST39 isolates, this lineage of *C. difficile* need to be further noticed. As some previous descriptions were too strong, we have rewritten the words in previous lines 338-339 as “Considering the high detection rate and antibiotic resistance rate of ST39 isolates, more attention should be paid on the molecular epidemiology and antibiotic resistance surveillance, in case the outbreak of *C. difficile* ST39 in this region” (lines 351-353).

6. Lines 343-345 doesn't make sense—"which was the hidden danger of hospital transmissions"?
Consider re-phrasing.

Reply: Thank you for the reviewer's suggestion. We have changed the "**which was the hidden danger of hospital transmissions**" to "which was the potential risk of *C. difficile* in-hospital transmission" (lines 367-368).

7. Lines 353-356 – how do these stated limitations affect the interpretations in this study?

Reply: Thank you for the question. We have added how the limitations affect this study as " In this study, only patients were included for *C. difficile* screening, which could explain the lower proportion of epidemiological linkage among the genetically-linked isolates" (lines 377-379).

Reviewer #2 (Public repository details (Required)):

Whole genome sequences should be deposited.

Reply: Thank you for the reminder. We have uploaded all the assembly genome files sequenced in our research to NCBI database under BioProject PRJNA686004.

Reviewer #2 (Comments for the Author):

Wen and colleagues present a genomic epidemiology analysis of C. difficile colonization in a large medical center in China. A total of 953 stool samples were screened over the 1-year study period, with C. difficile cultured from 173. In addition, 106 isolates from the prior 4 years were included. Genomic analysis revealed the following key results: 1) the diversity of sequence types is quite different than what has been described in the U.S. and Europe, with a dominance of clade 4, 2) the colonization rate was high - with ~20% of patients harboring C. difficile, 3) both toxin and non-toxin producing strains were identified, and had differences in their antibiotic resistance profiles, 4) extensive genomic linkages were identified between patients, suggesting significant transmission was occurring and 5) almost 50% of colonized individuals were deemed to have developed C. difficile infection (CDI). Overall, this is a very nice study, with an impressive sample size, solid analysis and intriguing results. My comments are largely to provide more details regarding the study design and add additional analyses to expand on and/or provide context to some of the more provocative findings.

Thank you for the carefully reading of our manuscript and the approval to our work. However, on account of our unclear expression, there is a misunderstanding for the fifth point “*almost 50% of colonized individuals were deemed to have developed C. difficile infection (CDI)*”. We have explained it in the 4th comment’s reply and revised our manuscript carefully in accordance with the following

comments.

Major comments

1. More details should be provided on the sample collection strategy. In particular: 1) how were patients identified for screening? 2) what fraction of patients were screened? and was there variability in the comprehensiveness of screening among wards or over time?

Reply: Thanks for your question. The samples in our study were collected from the clinical laboratory, and all the fecal samples sending for pathogens inspection were included for *C. difficile* screening. On account of this inspecting purpose, one third of the patients were from the division of gastroenterology. The isolation rates were diverse among different wards, which was higher in the division of gastroenterology, hematology and nephrology, and was lower in the division of pulmonary (the new Figure S4). We considered that could be associated with the patients who were going different underlying diseases, such as malignancy, IBD, nephrosis and pneumonia (the new Figure S2c and d). Fluctuation was also shown on the isolation rate between different months (the new Figure S1). We considered this kind of fluctuation could be due to the different sample sizes between different months.

2. Overall, the author's find a significantly larger fraction of genomic linkages than has been reported in comparable studies of C. difficile colonization and/or infection. While this could be due to differences in local strain types or epidemiology, some additional analyses to confirm robustness of these findings are warranted. Studies in both Europe (Eyre et al., Clinical Infectious Diseases, 2018) and the U.S.(Miles-Jay et al., Microbial Genomics, 2021) have found that variant distances can be misleading for certain strain types, as identical strains can be observed across large temporal or geographic distances. To provide additional context to the detected genetic linkages, I would advise

incorporating in the analysis publicly available genomes from high-prevalence STs, preferable ones that have been sequenced in China. In addition, plots showing how genetic distance relates to the time between isolates would also provide a sense of how likely direct or indirect transmission is to account for genetic linkages among hospitalized patients.

Reply: Thank you for the insightful comments. First, we apologized to correct a mistake. After a careful review of our manuscript, we found an error in calculating the proportion of the genetically-linked cases, which was supposed to be “43.8% (110/251) of isolates being genetically-linked to a prior case”, and we revised this result in line 59 and line 273. In our study, 43.8% (110/251) of isolates had no more than 3 SNVs and 35.5% (89/251) of isolates had no more than 2 SNVs from at least one previous case (the new Figure 4A). It is consistent with the study (*Eyre et al., N Engl J Med, 2011*) which found 35% of the isolates had no more than 2 SNVs from at least one previous case. The reviewer said “*identical strains can be observed across large temporal or geographic distances*”, it is true and we had considered that. Identifying transmission cases just based on SNVs is not valid, so we combined the wards information and isolation date to analyse the nosocomial transmission. And for showing how likely direct or indirect transmission is to account for genetic linkages among hospitalized patients, we rebuilt a new Figure 5B to show the detailed transmission routes. After reading the two articles mentioned by the reviewer (*Eyre et al., Clinical Infectious Diseases, 2018* and *Miles-Jay et al., Microbial Genomics, 2021*), we performed the analysis of comparing the pairwise SNVs from the same STs (the new Figure S8). We found the most prevalent types in this study, ST39 and ST54, had the lower pairwise SNVs than other STs, probably explaining the larger fraction of genomic linkages. As the publicly available genomes in China is scarce, we couldn't compare the pairwise SNVs in the same ST effectively. We appreciate more researchers in China could perform *C. difficile* sequencing for better clarifying the molecular characteristics and transmission routes of *C. difficile* in this region.

3. To provide context for the patient pairs deemed to be linked by transmission, it would be helpful to show histograms of genetic distances among isolate pairs from the same sequence type. This will give a sense of how much standing diversity there was.

Reply: Thanks for your suggestion. We have added a new Figure S8 in this manuscript which showed the histograms of genetic distances among isolate pairs from the same sequence type (lines 358-362).

4. The author's report a shockingly high rate of infection among colonized cases (45%). As this is more than an order of magnitude larger than previous estimates, more details should be reported. In particular: 1) how was the definition of clinical CDI different than prior reports of infection incidence among C. difficile carriers? 2) were C. difficile infections restricted to toxigenic strains, or did they also include non-tox? 3) were there any clinical variables associated with infection? 4) were any STs associated with infection? Also, added discussion of possible sources of this finding would be valuable.

Reply: I am sorry that this part has made some misunderstandings in the original manuscript. For better expression, we abbreviated *C. difficile* asymptomatic colonized patients as CDAC. Previously, researchers considered only CDI patients played a role on nosocomial transmission, while some subsequent jobs had emphasized the role of CDAC patients. In our research, not all the patients carrying *C. difficile* have the diarrhoea symptom. To figure out who have the stronger ability to spread *C. difficile*, we defined patients as CDI or CDAC based on with or without clinically significant diarrhoea (≥ 3 times per 24 h or unshaped) on the sampling date. The result showed that of the 110 isolates genetically-linked to a prior case, 50% of the isolates were associated with CDI cases, whereas 50% of the isolates were associated with CDAC cases, indicating CDI and CDAC patients had the same potential to cause *C.*

difficile spreading. For better understanding this result, we have revised the contents of this part in lines 296-301 and lines 364-366.

Minor comments

1. In Table 1 the p -values should be adjusted to a standard number of significant figures

Reply: Thank you for the reviewer's suggestion. We have adjusted the p -values to a standard number of significant figures in Table 1.

November 12, 2021

Prof. Guo-Bao Tian
Sun Yat-sen University
Zhongshan School of Medicine
Zhongshan School of Medicine, Sun Yat-sen University
74 Zhongshan 2nd Road
Guangzhou, China 510080
China

Re: Spectrum01322-21R1 (Whole-genome sequencing reveals high nosocomial transmission and serious antimicrobial resistance of *Clostridioides difficile* in a single centre in China, a four-year retrospective study)

Dear Prof. Guo-Bao Tian:

Thank you for submitting your manuscript to Microbiology Spectrum. As you will see your paper is very close to acceptance. Please modify the manuscript along the lines I have recommended below. As these revisions are quite minor, I expect that you should be able to turn in the revised paper in less than 30 days, if not sooner. If your manuscript was reviewed, you will find the reviewers' comments below.

Please have the manuscript edited for English usage. There is a list of services that the ASM maintains at <https://journals.asm.org/content/language-editing-services> - this is a list provided for your convenience. In particular, there are multiple errors in tense and usage:

Line 57 - none should be no, 61 with should be as, 69 delete "a", Line 71 - "the diversity", also remove reference to "serious" resistance to address the comment of the reviewer, 169 "the", 182 - are not were, 183 rate not rates, 253 - were not was, 254 - in not for, etc. This is not a complete or exhaustive list, but these errors were the ones I noticed in reviewing your revisions.

When submitting the revised version of your paper, please provide (1) point-by-point responses to the issues I raised in your cover letter, and (2) a PDF file that indicates the changes from the original submission (by highlighting or underlining the changes) as file type "Marked Up Manuscript - For Review Only". Please use this link to submit your revised manuscript. Detailed instructions on submitting your revised paper are below.

Link Not Available

Sincerely,

S. Wesley Long

Reviewer comments:

Preparing Revision Guidelines

- point-by-point responses to the issues I raised in your cover letter

- Upload a compare copy of the manuscript (without figures) as a "Marked-Up Manuscript" file.
- Each figure must be uploaded as a separate file, and any multipanel figures must be assembled into one file.
- Manuscript: A .DOC version of the revised manuscript
- Figures: Editable, high-resolution, individual figure files are required at revision, TIFF or EPS files are preferred

Please return the manuscript within 60 days; if you cannot complete the modification within this time period, please contact me. If you do not wish to modify the manuscript and prefer to submit it to another journal, please notify me of your decision immediately so that the manuscript may be formally withdrawn from consideration by Microbiology Spectrum.

We appreciated that you pointed the errors in tense and usage in our manuscript, and we apologized for not checking it carefully. We asked International Science Editing (<http://www.internationalscienceediting.com>) for helping editing this manuscript, and we believed that there were few errors in the final version now.

November 23, 2021

Prof. Guo-Bao Tian
Sun Yat-sen University
Zhongshan School of Medicine
Zhongshan School of Medicine, Sun Yat-sen University
74 Zhongshan 2nd Road
Guangzhou, China 510080
China

Re: Spectrum01322-21R2 (Whole-genome sequencing reveals the high nosocomial transmission and antimicrobial resistance of *Clostridioides difficile* in a single centre in China, a four-year retrospective study)

Dear Prof. Guo-Bao Tian:

Your manuscript has been accepted, and I am forwarding it to the ASM Journals Department for publication. You will be notified when your proofs are ready to be viewed.

Sincerely,

S. Wesley Long
Editor, Microbiology Spectrum

Journals Department
Supplemental files: Accept